# Brochoscopic Airway Clearance Therapy vs. Conventional Sputum Aspiration: The Future of Flexible Brochoscopes in Intensive Care Units?

**DOI:** 10.3390/diagnostics13203276

**Published:** 2023-10-22

**Authors:** Anjie Yao, Zixuan Liu, Wenni He, Hanyu Rao, Changhui Wang, Shuanshuan Xie

**Affiliations:** 1Department of Respiratory Medicine, Shanghai Tenth People’s Hospital, School of Medicine, Tongji University, Shanghai 200072, China; yannj2133279@163.com; 2School of Medicine, Tongji University, Shanghai 200092, China; 1950327@tongji.edu.cn (Z.L.); 1951187@tongji.edu.cn (W.H.); hanyu_rao@126.com (H.R.)

**Keywords:** flexible bronchoscopes, airway clearance therapy, severe pneumonia, invasive mechanical ventilation, intensive care unit, survival to hospital discharge

## Abstract

(1) Background: The aim of our study is to investigate the effectiveness of bronchoscopic airway clearance therapy (B-ACT) on severe pneumonia (SP) patients with invasive mechanical ventilation (IMV) in the intensive care unit (ICU). (2) Methods: Our study retrospectively enrolled 49 patients with sputum aspiration and 99 patients with B-ACT, and the latter were divided into the ≤once every 3 days group (*n* = 50) and >once every 3 days group (*n* = 49). (3) Results: We found most laboratory blood results were significantly improved in the B-ACT group as compared with those in sputum aspiration group (*p* < 0.05). Patients in the B-ACT group and those in ≤once every 3 days group also had significantly better survival to hospital discharge than those in their counterpart groups (Logrank *p* < 0.001). In patients with cardiopulmonary diseases or positive cultures for bacteria, the B-ACT group and those in the ≤once every 3 days group had significantly better survival outcomes to discharge than those in their counterpart groups (Logrank *p* < 0.001). B-ACT and the average frequency of ≤once every 3 days had significantly better impact on survival outcomes than their counterpart groups (HR: 0.444, 95% CI: 0.238–0.829, *p* = 0.011; HR: 0.285, 95% CI: 0163–0.498, *p* < 0.001). (4) Conclusions: In the future, flexible bronchoscopes may paly an important role in ACT for SP patients with IMV.

## 1. Introduction

Among all infectious diseases, pneumonia is the leading cause of death for all age groups worldwide [1]. Pneumonia patients with acute respiratory failure requiring endotracheal intubation and invasive mechanical ventilation (IMV) can be diagnosed with severe pneumonia (SP), for whom direct admission to an intensive care unit (ICU) is strongly recommended [2]. The mortality rate of SP is as high as 30–50% in the ICU [3]. Compared with patients with non-invasive ventilation, SP patients receiving IMV have more deposition of mucus and microbial secretions mainly because of biofilm formation in the endotracheal tube and medical device-related infections, which may lead to a decline in airway patency, and even aggravate the lung inflammation after long-term ventilation [4]. Therefore, it is very important to improve the airway clearance therapy (ACT) of SP patients during IMV in ICU [5].

Clinically, conventional ACT mainly includes: cough or breathing exercise, auxiliary positive pressure ventilation, chest wall oscillation sputum expulsion, artificial sputum aspiration, and postural drainage [6]. Bronchoscopic airway clearance therapy (B-ACT) is a new technology which inserts the flexible bronchoscope into the primary bronchus and the subsegmental bronchi to perform artificial sputum aspiration and even give alveolar lavage for the sake of clearing airway secretions and reducing pulmonary inflammation [7]. Considering the pandemic of COVID-19 or other infectious diseases recently and the risk of pathogen transmission, B-ACT is increasingly widely used in patients with pulmonary infectious diseases in recent years [8]. Infectious Diseases Society of America/American Thoracic Society consensus guidelines highlight that further trials are urgently needed to evaluate the effects of ICU care, especially the ACT, in SP patients with IMV [9]. Currently, the Respiratory Endoscopy Committee of the Chinese Medical Association also recommended the appropriate application of flexible bronchoscopes in critically ill patients in the ICU [10]. However, the effectiveness of B-ACT in SP patients during IMV remains controversial over the years, knowing that B-ACT recommendations in currently available guidelines mainly rely on studies enrolling SP patients without undergoing IMV [11,12]. In addition, to the best of our knowledge, there are few studies reporting the optimal frequency of B-ACT and the target groups patients that benefit most from it in SP patients undergoing IMV, especially in the ICU.

The aim of our study was to explore the benefits and risks of B-ACT in SP patients undergoing IMV in the ICU. We also want to determine the mean optimal frequency of B-ACT to some extent and special patient populations that benefit most from this new technology, knowing that it will have a positive impact on promoting the clinical application of flexible bronchoscopes and improving airway management of these critically ill patients in future.

## 2. Materials and Methods

### 2.1. Patients

Initially enrolled in this retrospective study were 868 patients diagnosed with SP during hospitalization in Shanghai Tenth People’s Hospital (Shanghai, China) from January 2020 to December 2022, of whom 230 SP patients undergoing IMV because of severe pneumonia met the major criteria of endotracheal intubation and IMV specified by the Infectious Diseases Society of America/American Thoracic Society consensus guidelines and were included for analysis [2]. We continued to screen these patients according to our exclusion criteria: (I) no ACT performed during hospitalization; (II) incomplete medical records and orders related to bronchoscopy; and (III) bronchoscopy for reasons other than airway clearance (Figure 1). Finally, 99 patients with B-ACT and 49 patients with artificial sputum aspiration were enrolled in this study.

### 2.2. Treatment Methods

In our study, the principles and procedures of bronchoscopy for patients closely followed the Chinese expert consensus on the clinical application of single-use (disposable) flexible bronchoscopes [10]. SP patients in the non-bronchoscopy group were given artificial sputum aspiration for ACT based on the systemic anti-infective and symptomatic supportive therapies according to other guidelines [9]. Artificial sputum suction was conducted by negative pressure suction under IMV. SP patients in bronchoscopy group received IMV using the flexible bronchoscopes for ACT based on the anti-infective and supportive treatments. The flexible bronchoscope clinically consists of single-use flexible bronchoscopes (SUFB) and reusable flexible bronchoscopes (RFB). Most of the flexible bronchoscopes used in this study were SUFB called H-Steriscopes (Vathin Medical Instrument Co., Ltd., Hunan, China), and the rest were RFB called V-bronchoscopes (Seesheen Medical Instrument Co., Ltd., Guangdong, China). The operation steps are as follows: Firstly, the comprehensive condition of the patients was assessed to decide whether they were able to tolerate B-ACT. Preoperative discussion and analysis were conducted according to the surgical requirements. After being fully informed of the procedure, the patients or their relatives decided to accept bronchoscopy. Secondly, patients were laid supine and sedated under continuous IMV. One side of the disposable sputum cup was connected to the bronchoscope, and the other side was connected to the negative pressure. The bronchoscope was inserted into the endotracheal tube to aspirate the visible secretions under direct version of the bronchoscope. Vital signs including oxygen saturation were monitored throughout the procedure. If necessary, 100–250 mL normal saline was used for bronchoalveolar lavage according to the patient condition. Finally, the airway condition was checked carefully before withdrawal. Airway secretion microbial cultures and drug sensitive tests were routinely carried out after bronchoscopy. Surgical records, medical records and instructions should be improved. The preoperative disease evaluation and postoperative close observation were also important to the patients.

### 2.3. Data Collection

Due to the retrospective nature of our study, all patients’ electronic files including physician’s order sheets, nurse’s recording sheets, blood laboratory results, lung imaging tests and daily medical records were rechecked and recorded carefully. The primary outcome measure was survival to hospital discharge. The observation indicators of our study included: (i) basic characteristics: sex, age, body mass index (BMI), length of hospitalization, duration of IMV, records of tracheotomy and outcomes of hospital discharge; (ii) bronchoscopy records: specific time, number and mean frequency; (iii) laboratory results: 3–5 mL fasting cubital venous blood and arterial blood were collected from patients of the two groups 1 day before and after the whole course of B-ACT, and the blood samples were analyzed by the specialized equipment of the hospital. The laboratory results included routine blood parameters, blood gas analysis, inflammatory factors, blood biochemical parameters, and myocardium markers.

### 2.4. Statistical Analysis

All analyses were performed using SPSS (version 25.0) and GraphPad Prism (version 8.0) software. The Kolmogorov–Smirnov test and Levene test were used for identifying the normality and homogeneity of variables. Categorical variables are presented as frequencies and percentages, while continuous variables are presented as the mean ± standard deviation (SD) for normally distributed variables, median and interquartile range for non-normally distributed variables. An independent group *t*-test was applied for normally distributed variables, while Mann–Whitney U test was used for non-normally distributed variables. Categorical variables were analyzed by the Chi-square test or Fisher’s exact test. The univariate and multivariate COX regression analysis was used to assess independent risk factors for the outcomes to hospital discharge. The difference of survival to hospital discharge between different groups were compared by the Kaplan–Meier analysis. *p* < 0.05 was considered statistically significant.

## 3. Results

### 3.1. Baseline Cohort Characteristics

There were no significant differences in the baseline characteristics including age, BMI, gender and commodities between the bronchoscopy and non-bronchoscopy groups (all *p* > 0.05) (Table 1). Next, patients in the bronchoscopy group were divided into ≤once every 3 days group (*n* = 50) and >once every 3 days group (*n* = 49) according to the mean frequency of bronchoscopy (mean frequency of bronchoscopy = days of bronchoscopy/numbers of bronchoscopies). The general characteristics of the patients were also similar in terms of age, BMI, gender and commodities between the two groups (all *p* > 0.05) (Table 1).

### 3.2. Preoperative and Postoperative Blood Laboratory Results

There were no significant differences in all routine blood parameters, blood gas analysis, inflammatory factors, blood biochemical parameters and myocardium markers before bronchoscopy between the bronchoscopy and non-bronchoscopy groups (all *p* > 0.05). There was significant improvement in most of the above blood laboratory results in the same group before and after B-ACT (*p* < 0.05) (Table 2; see Appendix A). Most importantly, the number of WBC (*p* < 0.001), percentage of neutrophils (N%; *p* = 0.001), the levels of arterial oxygen pressure (PO_2_; *p* = 0.005), oxygen saturation (SO_2_; *p* < 0.001), procalcitonin (PCT; *p* < 0.001) and C-reactive protein (CRP; *p* = 0.005) were all significantly improved in the bronchoscopy group as compared with those in the non-bronchoscopy group after B-ACT (Table 2). In addition, the levels of blood glucose (Glu; *p* < 0.001), urea (*p* = 0.037), creatinine (Cr; *p* < 0.001), albumin (Alb; *p* < 0.001), troponin I (TnI; *p* < 0.001), brain natriuretic peptide (BNP; *p* < 0.001) and D-dimer (*p* < 0.001) in the bronchoscopy group were also significantly better than those in the non-bronchoscopy group after B-ACT (see Appendix A). The above analysis indicated that B-ACT could help improve the function of multiple organs including lung, heart and kidney.

### 3.3. Comparison of Changes of Important Indicators

Furthermore, the violin plots showed that the changes of PCT (−5.43 vs. −3.96, *p*= 0.0059; Figure 2A) and CRP value (−109.2 vs. −73.99, *p* = 0.0198; Figure 2B) in the bronchoscopy group were more pronounced than those in the non-bronchoscopy group. The decline range of WBC (−6.24 vs. −3, *p* = 0.015; Figure 2D) and N% value (−16 vs. −6.2, *p* = 0.0004; Figure 2C) in the bronchoscopy group were larger than that in the non-bronchoscopy group. Also, the improvement of PO_2_ (19 vs. 9.6, *p* = 0.0108; Figure 2E) and SO_2_ (26.8 vs. 16.8; Figure 2F) in the bronchoscopy group was more remarkable than that in the non-bronchoscopy group.

### 3.4. The Kaplan–Meier Analysis and Survival Curves

Survival to hospital discharge of the patients in the bronchoscopy group was significantly better than that in the non-bronchoscopy group (Logrank *p* < 0.001; Figure 3A). The median survival time to discharge in the bronchoscopy group was significantly better than that in the non-bronchoscopy group (38 days vs. 17 days; see Appendix A). The mortality rate in the bronchoscopy group was significantly lower than that in the bronchoscopy group at day 14, 28 and 90 of hospitalization (60.5% vs. 91.7%; 24.3% vs. 71.9%; 4.9% vs. 17.1%) (see Appendix A). According to the mean frequency of bronchoscopy, we divided the patients of the bronchoscopy group into two groups: ≤once every 3 days and >once every 3 days. The survival curves showed that survival to discharge in the ≤once every 3 days group was significantly better than that in the > once every 3 days group (Logrank *p* < 0.001; Figure 3B). The median survival time to discharge also significantly better in the ≤once every 3 days group than that in the >once every 3 days group (53 days vs. 26 days; see Appendix A). The mortality rate in the ≤once every 3 days group was significantly lower than that in ≤once every 3-day group at day 14, 28 and 90 of hospitalization (89.1% vs. 94%;46.3% vs. 91.8%; 3.9% vs. 26.6%) (see Appendix A). Next, we divided them into various groups according to patients’ different characteristics. In patients with cardiopulmonary or cerebrovascular diseases, as well as patients with positive cultures for bacteria, we found that the bronchoscopy group had significantly better survival outcomes than non-bronchoscopy group (all Logrank *p* < 0.05; see Appendix A). Similarly, we also found that ≤once every 3 days group had significantly better survival rates than >once every 3 days group in patients with cardiopulmonary diseases, or positive cultures for bacteria (all Logrank *p* < 0.05; see Appendix A).

### 3.5. Univariate and Multivariate COX Regression Analysis

After adjusting for all covariates as confounding factors, the overall survival outcome to hospital discharge in the bronchoscopy group was significantly better than that in the non-bronchoscopy group (HR:0.444, 95% CI: 0.238–0.829, *p* = 0.011) (Table 3). Then, we divided the patients in the bronchoscopy group into two groups: ≤once every 3 days and >once every 3 days. After adjustment for all confounders, the overall survival outcome in the patients of the ≤once every 3 days group was significantly better than that in the once every 3 days group (HR: 0.285, 95% CI: 0.163–0.498, *p* < 0.001) (Table 4). In addition, the forest plots of HRs for survival to hospital discharge were generated to show the above analysis results between different groups more visually (Figure 4).

## 4. Discussion

In this study, we used flexible bronchoscopes to perform B-ACT in SP patients undergoing IMV. The results showed that the improvement of routine blood parameters (WBC# and N%), blood gas analysis (PO_2_, SO_2_) and inflammatory factors (PCT and CRP) in the bronchoscopy group was more pronounced than that in the non-bronchoscopy group. Survival to hospital discharge in the bronchoscopy group was significantly better than that in the non-bronchoscopy group. In addition, B-ACT had significantly positive impact on the survival to discharge of these patients. Especially in the patients with cardiopulmonary diseases or positive cultures for bacteria, B-ACT group and those in ≤once every 3 days group had pronounced better survival outcomes than those in their counterpart groups. Furthermore, we also explored the mean optimal frequency of bronchoscopy for SP patients undergoing B-ACT, and found that survival to discharge in the ≤once every 3 days group was significantly better than that in the >once every 3 days group, suggesting that more frequent bronchoscopy may to some extent be related to better prognosis of these patients. To summarize, B-ACT showed encouraging effects and advantages in various aspects including improvement of laboratory results and decrease in hospital morality, and average frequency of ≤once every 3 days may be a preferred mean optimal frequency of bronchoscopy for SP patients undergoing IMV.

Some previous studies have already demonstrated that B-ACT is significantly beneficial for children with SP [13,14], but its specific effectiveness for adults with SP has long been debated. Our analysis showed that B-ACT could improve some important laboratory results including some inflammatory factors (CRP and PCT), routine blood parameters (WBC# and N%) and blood gas analysis (PO_2_ and SO_2_). Consistent with our results, a retrospective study admitting 81 SP patients with IMV reported that not only inflammation indicators including tumor necrosis factor-a (TNF-α), CRP, IL-8 and IL-6 but also blood gas indexes including PO_2_ and SO_2_ were significantly better in the bronchoalveolar lavage (BAL) group than those in the control group after bronchoscopy, in which hospital stay, IMV time, and infection control window appearance time in the observation group were significantly shorter than those in the control group after bronchoscopy (all *p* < 0.05) [15]. Moreover, a prospective study enrolling 103 SP patients including patients with IMV demonstrated that the changes of APACHE II scores, inflammatory indicators (CRP, TNF-αand PCT) and blood gas analysis (oxygenation index (OI), PO_2_ and SO_2_) in the bronchoscopy group were more pronounced than those in the control group after treatment, and the length of stay in ICU, IMV time and duration of antibiotics of the bronchoscopy group were shorter than those of the control group after treatment (all *p* < 0.05) [16]. Consistently, a randomized controlled trial recruited 100 patients including patients with SP occurring within 48 h after tracheal intubation or 48 h after extubation, and demonstrated that the interleukin-8(IL-8), CRP, and PCT levels, as well as some respiratory mechanics indexes in the bronchoscopy group, were significantly lower than those in the control group after treatment (all *p* < 0.05) [17]. However, another retrospective case–control study that enrolled SP 72 patients including those with IMV concluded that BAL under bronchoscopy could increase PCT levels within 24 h, and markedly decreased PCT levels after 48 h (*p* < 0.05), which may indicate that more long-term clinical studies were urgently needed in the impacts of B-ACT on the different laboratory results in those patients after treatment [18].

Recently, the application of bedside B-ACT has been increasingly applied to airway management of SP patients undergoing IMV in the ICU. Our study indicated that B-ACT was significantly beneficial to the survival to hospital discharge of these patients. Consistently, a retrospective study divided 1560 with SP patients undergoing IMV into the bronchoscopy group and non-bronchoscopy group, and demonstrated that bronchoscopy during IMV was associated with reducing the risk of ICU (HR: 0.33, 95% CI:0.20–0.55, *p* < 0.001) and in-hospital mortality (HR: 0.40, 95% CI: 0.26–0.60, *p* < 0.001) in these patients [19]. However, very little research has investigated the optimal time and the number or frequency of B-ACT in SP patients undergoing IMV. It was found in our study that the mean bronchoscopy frequency of ≤once every 3 days may, to some extent, better improve the survival outcome to hospital discharge than that of >once every 3 days for SP patients during IMV. A retrospective cohort study found that the in-ICU mortality and 90-day mortality in the early bronchoscopy group (bronchoscopy within 24 h after intubation) were significantly lower than those in the late bronchoscopy group (in-ICU: 4.9% vs. 24.6%; 90-day: 11.8 vs. 32.8%), and was associated with lower 90-day mortality in multivariate analysis (HR:0.412, 95% CI: 0.192–0.883, *p* = 0.023), indicating that early bronchoscopy benefits the clinical outcome of mechanically ventilated patients with SP [20]. However, another retrospective observational study divided 229 SP children into the IMV group and non-IMV group, as well as the early BAL group (receiving BAL within 1 day of admission) and late BAL group, and found that early BAL under bronchoscopy could not improve the survival rate but can reduce the hospitalization and ICU time, exhibiting that B-ACT is still beneficial to the prognosis of SP patients [21]. Therefore, it is still necessary to have further prospective research to explore the appropriate number, time or frequency of B-ACT for SP patients undergoing IMV, knowing that more accurate, individualized and comprehensive airway clearance management is urgently required in such patients.

Furthermore, our analysis showed that more frequent bronchoscopy groups had better survival rates in hospital than control groups in SP patients with cardiopulmonary diseases or positive cultures for bacteria during IMV, which suggested that B-ACT may be more suitable for improving prognosis of these groups of patients. A previous study had demonstrated that B-ACT can improve the therapeutic effects on chronic obstructive pulmonary disease (COPD) patients complicated with SP during IMV, which can significantly reduce the ventilation time, hospital stay, reintubation rate and fatality rate, while increasing the weaning success rate [22]. However, some other studies demonstrated that beside B-ACT is also beneficial in improving pulmonary ventilation and reducing systemic inflammatory response of pneumonia patients with cerebral ischemic or hemorrhagic stroke [23,24,25]. In addition, polymerase chain reaction (PCR) or metagenomic next-generation sequencing (mNGS) based on the fluid of BAL after bronchoscopy can provide a higher identification rate of pathogens involved in pneumonia than direct examination and culture, which would hopefully improve early diagnosis and prognosis of critically ill patients, especially those with bacterium/fungus-associated pneumonia [26,27]. Therefore, further studies are needed to investigate the impacts of B-ACT on these critically ill patients with different genders, ages, regions or other characteristics, suggesting great value in clinical application [28].

As a new technology, B-ACT has both advantages and disadvantages. The biggest advantage of B-ACT is that flexible bronchoscopy could remove sputum and other secretions in deep airways effectively, which is beneficial to reducing further airway damage and lung inflammation by halting bacterial colonization [29]. Specifically, SUFB can replace sterile devices directly without disinfection, which not only reduces the spread of pathogens through aerosols and the risk of infection, but also makes it easier to sterilize and preserve other devices such as portable screens, thus reducing the procedures and costs of cleaning and storage in hospital [10,30,31]. RFB can provide better image quality, suction, maneuverability and medical record integration, which is more extensively used for advanced diagnostic and therapeutic procedures compared to SUFB [32]. After the current COVID-19 pandemic or other infectious diseases, it is more recommended to use SUFB in the airway management of patients with pulmonary infectious diseases than in the ICU [33]. However, over-frequent B-ACT may aggravate airway damage and lung inflammation [34]. The arterial partial pressure of oxygen could drop during bronchoscopy, which may increase the risk of respiratory failure in patients with mechanical ventilation [35]. Although the bronchoscopy procedure could be safely performed, some patients still have rare adverse events such as airway edema, submucosal hemorrhage, increased secretions, bronchospasm and even dyspnea after bronchoscopy [36]. Therefore, we should not only improve the safety and efficiency of flexible bronchoscopes, but explore the optimal operation of B-ACT including the appropriate time or number according to patients’ individual conditions in the future.

There are some limitations in our study. First, this study was a single-center retrospective study, not a randomized controlled trial, which may cause analysis bias. For example, the past original records did not include some bronchoscopy details such as postoperative complications, basic vital signs during bronchoscopy, immunology factors, pulmonary function tests and others. In addition, the 148 patients may not be able to represent all SP patients with IMV, leading to limited generalizability of conclusions. However, we discussed the specific impacts of B-ACT on the different laboratory results and the survival benefits of it in a relatively large number of SP patients with IMV for the first time. Most importantly, we compared different mean frequencies of flexible bronchoscopes affecting survival to hospital discharge of these patients, and identified appropriate group patients suitable for B-ACT, which suggested that more prospective cohort studies were needed to explore individualized and precise airway management for SP patients with IMV in future.

## 5. Conclusions

In this study, we demonstrated that B-ACT significantly prolonged the time of survival to hospital discharge and improved many different laboratory tests, which highlights the importance of flexible bronchoscopes in the airway management for SP patients undergoing IMV in the ICU. We also observed that the mean optimal frequency of flexible bronchoscopes affected the prognosis of such patients. It is our recommendation that the mean frequency of flexible bronchoscopes be ≤ once every 3 days in that it contributed to better survival outcomes than the mean frequency of >once every 3 days. And more frequent B-ACT may be more suitable for improving diagnosis in patients with cardiopulmonary diseases or positive cultures for bacteria. Hopefully, our study could provide a novel evidence-based individualized therapeutic reference for SP patients with IMV in the ICU.

## Figures and Tables

**Figure 1 diagnostics-13-03276-f001:**
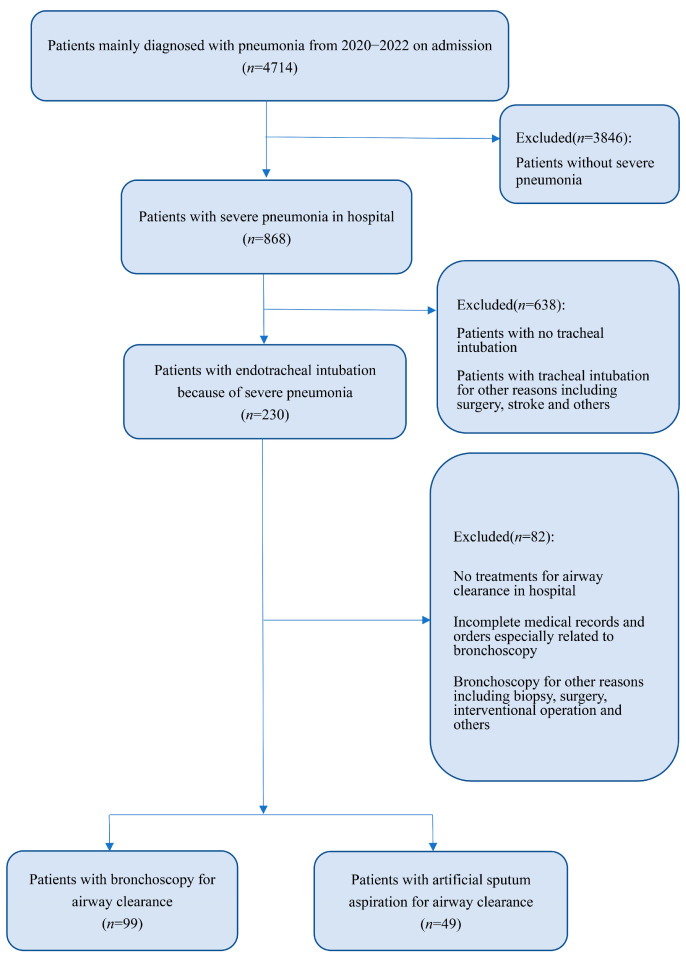
Flow diagram.

**Figure 2 diagnostics-13-03276-f002:**
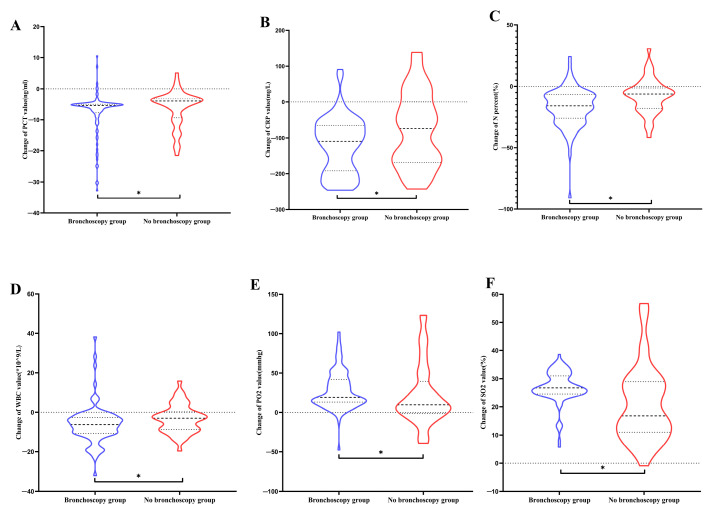
The violin plots of changes of important indicators between bronchoscopy and non-bronchoscopy groups. (**A**) Change in PCT value. (**B**) Change in CRP value. (**C**) Change in N%. (**D**) Change in WBC value. (**E**) Change in PO_2_ value. (**F**) Change in SO_2_ value. Abbreviations: PCT: procalcitonin; CRP: C-reactive protein; WBC: white blood cell; N: neutrophils; PO_2_: arterial oxygen pressure; SO_2_: oxygen saturation. The asterisk represented the significant differences.

**Figure 3 diagnostics-13-03276-f003:**
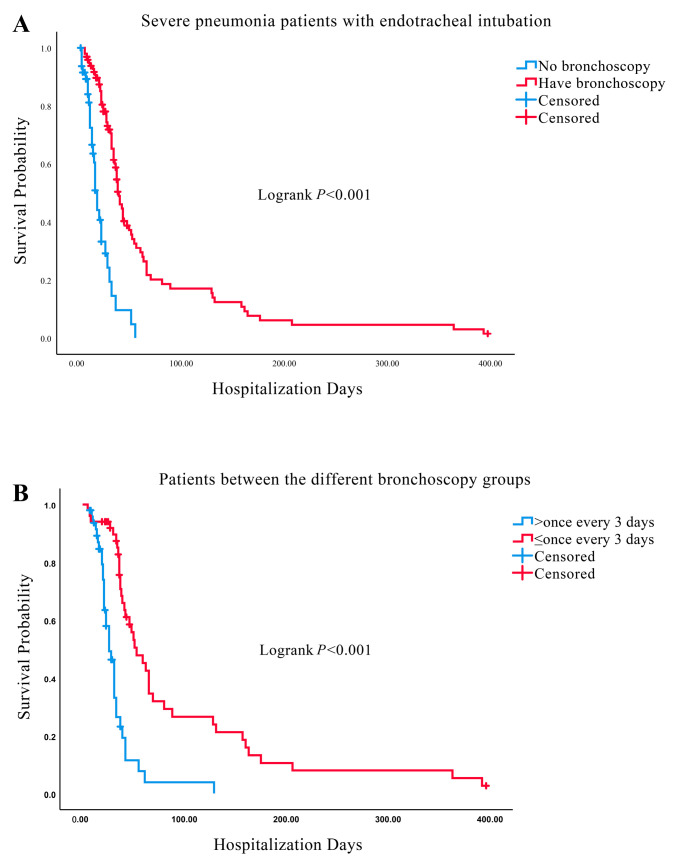
Survival curves between different groups. (**A**) Survival curves between bronchoscopy and non-bronchoscopy groups. (**B**) Survival curves between ≤once every 3 days and >once every 3 days groups.

**Figure 4 diagnostics-13-03276-f004:**
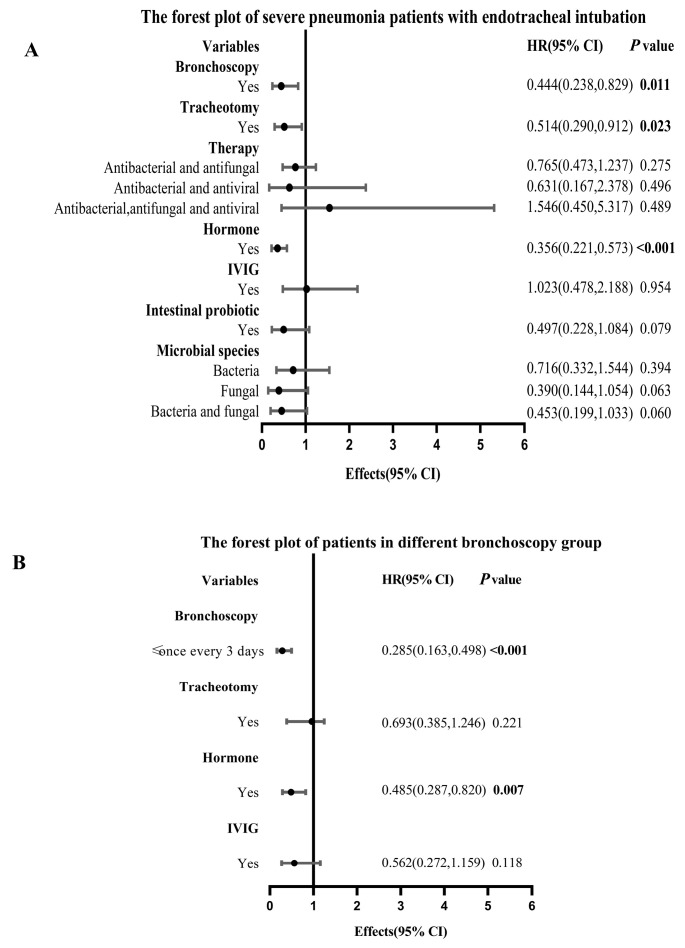
Forest plots between different groups. (**A**) The forest plot of SP patients undergoing bronchoscopy or not during IMV. (**B**) The forest plot of SP patients undergoing bronchoscopy during IMV. Abbreviations: SP: severe pneumonia; IMV: invasive mechanical ventilation; IVIG: intravenous immunoglobin; HR: hazard ratio; CI: confidence interval.

**Table 1 diagnostics-13-03276-t001:** Baseline characteristics of SP patients undergoing IMV between different groups.

Variables	Bronchoscopy *n* = 99	No Bronchoscopy *n* = 49	*p*	≤Once Every 3 Days *n* = 50	>Once Every 3 Days *n* = 49	*p*
Age	68(52, 76)	70(60.5, 78)	0.196	69(50.25, 76)	67(54, 76)	0.809
BMI	22.5817 ± 3.67431	23.7090 ± 3.70282	0.082	22.9947 ± 3.54403	22.1576 ± 3.79185	0.258
Sex			0.969			0.347
Male = 0	69(69.7%)	34(69.4%)		37(74%)	32(65.3%)	
Female = 1	30(30.3%)	15(30.6%)		13(26%)	17(34.7%)	
Comorbidities			0.393			0.985
No = 0	17(17.2%)	2(4.1%)		9(18%)	8(16.3%)	
HBP = 1	3(3.0%)	2(4.1%)		1(2%)	2(4.1%)	
CVA = 2	34(34.3%)	19(38.8%)		18(36%)	16(32.7%)	
CHD = 3	16(16.2%)	11(22.4%)		8(16%)	8(16.3%)	
DM = 4	10(10.1%)	5(10.2%)		5(10%)	5(10.2%)	
COPD = 5	13(13.1%)	6(12.2%)		7(14%)	6(12.2%)	
Others = 6	6(6.1%)	4(8.2%)		2(4%)	4(8.2%)	
Antibiotic therapy			0.958			0.946
B	35(35.4%)	16(33.3%)		18(36.0%)	17(34.7%)	
B + F	55(55.6%)	28(58.3%)		28(56.0%)	27(55.1%)	
B + V	3(3.0%)	2(4.2%)		1(2.0%)	2(4.1%)	
B + F+V	6(6.1%)	2(4.2%)		3(6.0%)	3(6.1%)	
Antibiotic days	18(5, 33)	12(6, 19)	0.082	19.5(11, 46)	15(3, 25)	0.109

Abbreviations: SP: severe pneumonia; IMV: invasive mechanical ventilation; BMI: body mass index; COPD: chronic obstructive pulmonary disease; CVA: cerebral vascular accident; CHD: coronary heart disease; DM: diabetes mellitus; HBP: high blood pressure.

**Table 2 diagnostics-13-03276-t002:** Comparisons of important laboratory results between bronchoscopy and non-bronchoscopy groups.

Laboratory Results	Have Bronchoscopy	No Bronchoscopy	*p* Value
Routine blood parameters			
PLT (10^9^/L)			
Pre treatment	185(113, 311)	172(121, 221.5)	0.315
Post treatment	202.9502 ± 109.45456	160(101.5, 249)	0.121
*p* value	0.224	0.423	
WBC (10^9^/L)			
Pre treatment	14.08(11.26, 18.61)	16.72(13.58, 19.485)	0.125
Post treatment	7.21(5.57, 9.48)	11.68(7.655, 15.975)	<0.001
*p* value	<0.001	0.002	
*N* (%)			
Pre treatment	91.8(88.8, 95.0)	92.2(88.55, 93.9)	0.347
Post treatment	74.8(65.6, 83.2)	84(73.6, 89.3)	0.001
*p* value	<0.001	<0.001	
Blood gas analysis			
PH			
Pre treatment	7.32(7.22, 7.40)	7.31(7.21, 7.37)	0.181
Post treatment	7.40(7.39, 7.42)	7.41(7.335, 7.4535)	0.382
*p* value	<0.001	<0.001	
PCO_2_ (mmHg)			
Pre treatment	42.8929 ± 12.15666	42.9(38.1, 49.95)	0.389
Post treatment	40(37, 43)	39(32.35, 45.95)	0.191
*p* value	0.076	0.009	
PO_2_ (mmHg)			
Pre treatment	66(56, 71)	68.7(59.4, 72.5)	0.167
Post treatment	84(77.7, 100)	74(68.4, 96.9)	0.005
*p* value	<0.001	0.009	
SO_2_ (mmHg)			
Pre treatment	62.2(58, 64.4)	65(49, 75.5)	0.225
Post treatment	89(88.9, 90)	83(79.05, 88.85)	<0.001
*p* value	<0.001	<0.001	
Inflammatory factors			
PCT (ng/mL)			
Pre treatment	6.12(5.26, 9.90)	5.80(4.915, 12.12)	0.189
Post treatment	0.52(0.11,2.90)	2.2(1.395, 4.180)	<0.001
*p* value	<0.001	<0.001	
CRP (mg/L)			
Pre treatment	168.23(109.18, 250)	160.6(80.33, 201.77)	0.220
Post treatment	25.36(8.80, 71.40)	67(17.895, 135.005)	0.005
*p* value	<0.001	<0.001	

Abbreviations: PLT: platelets; WBC: white blood cell; N: neutrophils; PH: pondus hydrogenii; PCO_2_: partial pressure of carbon dioxide; PO_2_: arterial oxygen pressure; SO_2_: oxygen saturation; PCT: procalcitonin; CRP: C-reactive protein.

**Table 3 diagnostics-13-03276-t003:** COX regression analysis of survival outcomes to hospital discharge in patients undergoing bronchoscopy or not.

Variables	Univariate Analysis		Multivariate Analysis	
	HR (95% CI)	*p* Value	HR (95% CI)	*p* Value
Bronchoscopy		<0.001		0.011
No	Reference		Reference	
Yes	0.247(0.154, 0.397)	<0.001	0.444(0.238, 0.829)	0.011
Tracheotomy		<0.001		0.023
No	Reference		Reference	
Yes	0.364(0.235, 0.565)	<0.001	0.514(0.290, 0.912)	0.023
Therapy		0.018		0.518
Antibacterial	Reference		Reference	
Antibacterial and antifungal	0.559(0.374, 0.835)	0.005	0.765(0.473, 1.237)	0.275
Antibacterial and antiviral	1.743(0.539, 5.631)	0.353	0.631(0.167, 2.378)	0.496
Antibacterial, antifungal and antiviral	0.677(0.210, 2.183)	0.513	1.546(0.450, 5.317)	0.489
Hormone		<0.001		<0.001
No	Reference		Reference	
Yes	0.363(0.241, 0.547)	<0.001	0.356(0.221, 0.573)	<0.001
IVIG		0.028		0.954
No	Reference		Reference	
Yes	0.492(0.261, 0.926)	0.028	1.023(0.478, 2.188)	0.954
Intestinal probiotic		0.012		0.079
No	Reference		Reference	
Yes	0.475(0.266, 0.847)	0.012	0.497(0.228, 1.084)	0.079
Microbial species		<0.001		0.130
No = 0	Reference		Reference	
Bacteria = 1	0.507(0.255, 1.010)	0.054	0.716(0.332, 1.544)	0.394
Fungal = 2	0.289(0.121, 0.693)	0.005	0.390(0.144, 1.054)	0.063
Bacteria and fungal = 3	0.240(0.120, 0.479)	<0.001	0.453(0.199, 1.033)	0.060

Abbreviations: HR: hazard ratio; CI: confidence interval; IVIG: intravenous immunoglobin.

**Table 4 diagnostics-13-03276-t004:** COX regression analysis of survival outcomes to hospital discharge in patients undergoing bronchoscopy.

Variables	Univariate Analysis		Multivariate Analysis	
	HR (95% CI)	*p* Value	HR (95% CI)	*p* Value
Bronchoscopy		<0.001		<0.001
>once every 3 days	Reference		Reference	
≤once every 3 days	0.259(0.154, 0.437)	<0.001	0.285(0.163, 0.498)	<0.001
Tracheotomy		0.013		0.221
No	Reference		Reference	
Yes	0.492(0.281, 0.859)	0.013	0.693(0.385, 1.246)	0.221
Therapy		0.480		
Anti-bacteria	Reference			
Anti-bacteria and anti-fungal	0.757(0.468, 1.225)	0.257		
Anti-bacteria and anti-viral	1.759(0.415, 7.461)	0.444		
Anti-bacteria, anti-fungal and anti-viral	1.155(0.346, 3.856)	0.814		
Hormone		0.001		0.007
No	Reference		Reference	
Yes	0.445(0.271, 0.732)	0.001	0.485(0.287, 0.820)	0.007
IVIG		0.040		0.118
No	Reference		Reference	
Yes	0.478(0.236, 0.968)	0.040	0.562(0.272, 1.159)	0.118
Intestinal probiotic		0.056		
No	Reference			
Yes	0.548(0.296, 1.015)	0.056		
Microbial species		0.087		
No = 0	Reference			
Bacteria = 1	0.597(0.203, 1.751)	0.347		
Fungal = 2	0.378(0.113, 1.269)	0.115		
Bacteria and fungal = 3	0.344(0.120, 0.985)	0.047		

Abbreviations: HR: hazard ratio; CI: confidence interval; IVIG: intravenous immunoglobin.

## Data Availability

All data generated or analyzed during current study are included in this published article, and are available from the corresponding author upon reasonable request.

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
