# Peer review of "Brochoscopic Airway Clearance Therapy vs. Conventional Sputum Aspiration: The Future of Flexible Brochoscopes in Intensive Care Units?"

_diagnostics, 2023, doi:10.3390/diagnostics13203276_

Round 1

Reviewer 1 Report

I have no additional suggestions. The article design is well done. The large number of cases increased the value of the study. Comparisons between groups are explained in detail.  The statistical methods used and their results are explained in detail. Tables that support the literature have been created in detail.  Necessary citations have been made in reference articles in the discussion.

Author Response

Response to Reviewer 1 Comments

Dear editors and reviewers,

Thank you for your careful review and valuable suggestions of our manuscript entitled “Brochoscopic airway clearance therapy vs. conventional sputum aspiration: the future of flexible brochoscopes in intensive care unit?”. The following are our point-by-point responses to reviewers and editors’ comments. The comments are shown in italics and are followed by our responses.

Editors:

Thank you very much for your work. I have modified the paper according to your requirements.

Reviewer 1:

I have no additional suggestions. The article design is well done. The large number of cases increased the value of the study. Comparisons between groups are explained in detail.  The statistical methods used and their results are explained in detail. Tables that support the literature have been created in detail.  Necessary citations have been made in reference articles in the discussion.

Response: Thank you very much for your affirmation. We have reviewed all the requirements and confirmed that our revision meet the requirements of the magazine. Thanks again for your efforts on our work.

Reviewer 2 Report

Very interesting article with practical impact

Author Response

Response to Reviewer 2 Comments

Dear editors and reviewers,

Thank you for your careful review and valuable suggestions of our manuscript entitled “Brochoscopic airway clearance therapy vs. conventional sputum aspiration: the future of flexible brochoscopes in intensive care unit?”. The following are our point-by-point responses to reviewers and editors’ comments. The comments are shown in italics and are followed by our responses.

Editors:

Thank you very much for your work. I have modified the paper according to your requirements.

Reviewer 2:

Very interesting article with practical impact.

Response: Thank you very much for your affirmation. We have reviewed all the requirements and confirmed that our revisions meet the requirements of the magazine. Thanks again for your efforts on our work.

Reviewer 3 Report

1. line 41 - you mentioned cough and breathing exercises as conventional ACT but we talk about IMV, please clarify

2. in the material and method section please specify more clear the inclusion criteria, including also the definition for severe pneumonia in your patients. 

3. the exclusion criteria from the diagram differ from the ones specified in the text, please explain (they are more complex in the diagram). 

4.  The mortality rate in the bronchoscopy group was significantly lower than that in the 201 bronchoscopy group at day 14, 28 and 90 of hospitalization (60.5% vs. 91.7%; 24.3% vs. 202 71.9%; 4.9% vs. 17.1%) - something wrong, please clarify

5. all patients had positive cultures for bacteria? if yes, when it was taken? 

6. in your study, the antibiotherapy was chosen based on microbiological culture? if yes, in the two groups it was any difference between them in the antibiotherapy length or choice of the antibiotic.

7. any of these patients had other pulmonary comorbidities? like lung fibrosis or bronchiectasis? maybe is important their history before the ICU admission.. how was treated the pneumonia? the number of days of hospitalisation before the ICU admission

8. how was taken the decision for B-ACT in ICU patients? based on which criteria? please specify

10. did you observe any complications after the B-ACT? If yes in how many patients?

11. the conclusion is too general, please consider rewriting it. 

Author Response

Reviewer 3:

Point 1: line 41 - you mentioned cough and breathing exercises as conventional ACT but we talk about IMV, please clarify

Response 1: Thank you for your constructive suggestions. We are very sorry for the mistake. The expression here was not rigorous enough. For pneumonia patients on general wards, the conventional ACT concludes cough and breathing exercises. But for SP patients with IMV in ICU, it’s impossible to accomplish cough and breathing exercises. We deleted “cough and breathing exercises” and replaced “clinically” by “for SP patients with IMV in the ICU” (please see line 41 in the modified version).

Point 2: in the material and method section please specify more clearly the inclusion criteria, including also the definition for severe pneumonia in your patients. 

Response 2: Thank you very much for your efforts on our work. Your suggestion is very to the point. We added “According to the Infectious Diseases Society of America and the Thoracic Society (please see lecture 1) [1], the diagnostic criteria of severe pneumonia should meet 1 of the following major criteria or at least 3 of the following minor criteria:(I) major criteria: (1) endotracheal intubation and invasive mechanical ventilation (IMV); (2) septic shock with the need of vasopressors. (II) minor criteria: (1) respiratory rate≥30 breaths/min; (2) arterial oxygen pressure/fraction of inspired oxygen (PaO2/FiO2 ratio) ≤250; (3) multilobar infiltrates; (4) confusion/disorientation; (5) blood urea nitrogen (BUN) level≥20 mg/dL; (6) white blood cell (WBC) count<4000 cells/mm3; (7) platelet count<100,000 cells/mm3; (8) core temperature<36℃; (9) hypotension requiring aggressive fluid resuscitation. In our study, the SP patients must meet the following major criteria: endotracheal intubation and IMV ” to the beginning of the paragraph (please see line 69 in modified version).

[1] Mandell, L.A.; Wunderink, R.G.; Anzueto, A.; Bartlett, J.G.; Campbell, G.D.; Dean, N.C.; Dowell, S.F.; File, T.M., Jr.; Musher, D.M.; Niederman, M.S.; et al. Infectious Diseases Society of America/American Thoracic Society consensus guidelines on the management of community-acquired pneumonia in adults. Clinical infectious diseases : an official publication of the Infectious Diseases Society of America 2007, 44 Suppl 2, S27-72, doi:10.1086/511159.

Point 3: the exclusion criteria from the diagram differ from the ones specified in the text, please explain (they are more complex in the diagram). 

Response 3: Thank you very much for your reminding. The explanation was not enough in the text. The advice you raised is very meaningful. We replaced the “of whom 230 SP patients undergoing IMV because of severe pneumonia met the major criteria of endotracheal intubation and IMV specified by the Infectious Diseases Society of America/American Thoracic Society consensus guidelines and included for analysis” as “after excluding 638 patients with no tracheal intubation or tracheal intubation for reasons other than severe pneumonia, we enrolled 230 patients with endotracheal intubation and invasive ventilation for severe pneumonia based on the Infectious Diseases Society of America/American Thoracic Society consensus guidelines” (please see line 71 in the modified version).

Point 4: The mortality rate in the bronchoscopy group was significantly lower than that in the 201 bronchoscopy group at day 14, 28 and 90 of hospitalization (60.5% vs. 91.7%; 24.3% vs. 202 71.9%; 4.9% vs. 17.1%) - something wrong, please clarify

Response 4: Thank you very much for your reminding. I am very sorry for this mistake. According to the survival curves, the “mortality” should be replaced by “survival” in supplementary table S2 (Please see the new supplementary table S2 in the modified version).

Supplementary table S2. Median survival days and 14,28,90-day of survival rates in hospital between different groups.

Median survival days

14-day of survival rate (%)

28-day of survival rate (%)

90-day of survival rate (%)

Have bronchoscopy

38

91.7

71.9

17.1

No bronchoscopy

17

60.5

24.3

4.9

≤once every 3 days

53

94

91.8

26.6

>once every 3 days

26

89.1

46.3

3.9

Point 5: all patients had positive cultures for bacteria? if yes, when it was taken? 

Response 5: Thank you for your constructive suggestions. The advice you raised is very meaningful. Not all patients had positive cultures for bacteria. In bronchoscopy group (n=99), there were 5 patients with negative cultures, 30 patients with positive cultures for bacteria, 10 patients with positive cultures for fungal, and 54 patients with positive cultures for bacteria and fungal. In non-bronchoscopy group (n=49), there were 16 patients with negative cultures, 15 patients with positive cultures for bacteria, 5 patients with positive cultures for fungal, and 13 patients with positive cultures for bacteria and fungal. Then we had COX regression analysis based on these statistics (please see table 3 and table 4 in the modified version). For patients in our study, sputum samples were collected by artificial sputum suction tubes or flexible bronchoscopes during 24-72 hours after hospitalization. Sputum bacterial cultures and drug sensitive tests were routinely carried out immediately after each sample collection. We added expression of specific sample collection time to the treatment methods section (please see line 85 in modified version).

Point 6: in your study, the antibiotic therapy was chosen based on microbiological culture? if yes, in the two groups it was any difference between them in the antibiotic therapy length or choice of the antibiotic.

Response 6: Thank you for your constructive suggestions. Your suggestion is very to the point. During early hospitalization, the empirical antibiotic therapy was chosen to control infection timely. After results of microbiological cultures and drug sensitive tests coming out, the antibiotic therapy was chosen based on these results, as well as other recent laboratory results of patients. There was no significant difference between the two groups in the antibiotic therapy length or choice of antibiotic after statistical analysis. And the antibiotic therapy length or choice of antibiotic had no significant impact on prognosis of patients in different groups (please see table 3 and 4 in the modified version). We added related statistical analysis to the table 1 (please see modified table 1 in the modified version).

Table 1. Baseline characteristics of patients with SP undergoing IMV between different groups.

Variables

Bronchoscopy

n=99

No bronchoscopy

n=49

P

≤once every 3 days

n=50

>once every 3 days

n=49

P

Age

68(52,76)

70(60.5,78)

0.196

69(50.25,76)

67(54,76)

0.809

BMI

22.5817±3.67431

23.7090±3.70282

0.082

22.9947±3.54403

22.1576±3.79185

0.258

Sex

Male=0

 Female=1

69(69.7%)

30(30.3%)

34(69.4%)

15(30.6%)

0.969

37(74%)

13(26%)

32(65.3%)

17(34.7%)

0.347

Comorbidities

No=0

HBP=1

CVA=2

CHD=3

 DM=4

 COPD=5

 Others=6

Antibiotic therapy

B

 B+F

 B+V

 B+F+V

Antibiotic days

17(17.2%)

 3(3.0%)

34(34.3%)

16(16.2%)

10(10.1%)

13(13.1%)

 6(6.1%)

35(35.4%)

55(55.6%)

 3(3.0%)

 6(6.1%)

18(5,33)

 2(4.1%)

 2(4.1%)

19(38.8%)

11(22.4%)

 5(10.2%)

 6(12.2%)

 4(8.2%)

16(33.3%)

28(58.3%)

2(4.2%)

2(4.2%)

12(6,19)

0.393

0.958

0.082

9(18%)

1(2%)

18(36%)

8(16%)

5(10%)

7(14%)

2(4%)

18(36.0%)

28(56.0%)

 1(2.0%)

 3(6.0%)

19.5(11,46)

8(16.3%)

2(4.1%)

16(32.7%)

8(16.3%)

5(10.2%)

6(12.2%)

4(8.2%)

17(34.7%)         

27(55.1%)

 2(4.1%)

 3(6.1%)

15(3,25)

0.985

0.946

0.109

Abbreviations: SP: severe pneumonia; IMV: invasive mechanical ventilation; BMI: body mass index; COPD: chronic obstructive pulmonary disease; CVA: cerebral vascular accident; CHD: coronary heart disease; DM: diabetes mellitus; HBP: high blood pressure; B: antibacterial; F: antifungal; V: antiviral.

Point 7: any of these patients had other pulmonary comorbidities? like lung fibrosis or bronchiectasis? maybe is important their history before the ICU admission. how was treated the pneumonia? the number of days of hospitalization before the ICU admission

Response 7: Thank you very much for your meaningful advice. This is a very constructive suggestion to improve the quality of the research. There were 6 patients with other comorbidities in bronchoscopy group (2 patients with bronchiectasis, 1 patient with lung fibrosis, 1 patient with lung cancer, 1 patient with hypothyroidism and 1 patient with rheumatoid arthritis) and 4 patients with other comorbidities (1 patients with bronchiectasis, 1 patient with pulmonary emphysema, 1 patient with breast cancer and 1 patient with chronic nephritis) in non-bronchoscopy group. But there was no significant difference between the two groups in the types of comorbidities (please see table 1 in the modified version). Before the ICU admission, these pneumonia patients were mainly treated with antibiotics based on the microbiological cultures and drug sensitive test from sputum samples, and other routine symptomatic relief and supportive treatment in emergency room or general ward. There were different number of days of hospitalization before the ICU admission in these patients. But they were all transferred to ICU due to severity and complexity of the illness after that. Our study was focused on the treatments of these patients after the ICU admission. Before further analysis, we confirmed that all the base line characteristics between different groups had no significant difference (please see table 1 in the modified version), and all the laboratory results also had no significant difference between the groups (please see table 2 and supplementary table S in the modified version). By this way, we try to avoid the impact of treatments and condition of patients before the ICU admission on our study.

Point 8: how was taken the decision for B-ACT in ICU patients? based on which criteria? please specify

Response 8: Thank you very much for your efforts on our work. The advice you raised is very meaningful. According to Respiratory Endoscopy Committee of the Chinese Medical Association and other international associations (please see lecture 2, 3, 4) [2-4], the criteria for B-ACT in ICU patients was as follows: (1) For infectious diseases, unexplained pneumonia, and multi-drug resistant organisms or other severe diseases, B-ACT is recommended; (2) B-ACT can be used immediately for the management of deep airway clearance, difficult airways, airway foreign bodies, hemoptysis and other acute diseases in emergency situations; (3) B-ACT can be used to collect deep airway secretion for precise detection, which is important to diagnosis and treatment strategies of critically ill patients. We added related criteria to the “materials and methods” of manuscript (please see line 77 in the modified version).

  1. [Expert consensus on the clinical application of single-use (disposable) flexible bronchoscopes]. Zhonghua jie he he hu xi za zhi = Zhonghua jiehe he huxi zazhi = Chinese journal of tuberculosis and respiratory diseases 2023, 46, 977-984, doi:10.3760/cma.j.cn112147-20230519-00252.
  2. Solidoro, P.; Corbetta, L.; Patrucco, F.; Sorbello, M.; Piccioni, F.; D'Amato, L.; Renda, T.; Petrini, F. Competences in bronchoscopy for Intensive Care Unit, anesthesiology, thoracic surgery and lung transplantation. Panminerva medica 2019, 61, 367-385, doi:10.23736/s0031-0808.18.03565-6.
  3. Ho, E.; Wagh, A.; Hogarth, K.; Murgu, S. Single-Use and Reusable Flexible Bronchoscopes in Pulmonary and Critical Care Medicine. Diagnostics (Basel, Switzerland) 2022, 12, doi:10.3390/diagnostics12010174.

Point 9: did you observe any complications after the B-ACT? If yes in how many patients?

Response 9: Thank you very much for your efforts on our work. The advice you raised is very meaningful. Due to the retrospective nature of our study, we only observed some mild complications in very few patients (n=8) after B-ACT in the medical records, such as transient bronchospasm, mild airway edema, minor submucosal hemorrhage and transient hypoxia. After immediately symptomatic treatments, all the above complications were solved. Therefore, B-ACT is still a safe and effective treatment strategy for critically ill patients in the ICU.

Point 10: the conclusion is too general, please consider rewriting it. 

Response 10: Thank you for your constructive suggestions. Your suggestion is very to the point. I modified the conclusion as “In this study, we demonstrated that B-ACT significantly prolonged the time of improved survival to hospital discharge and improved many different laboratory parameters (PCT, CRP, WBC, N, PO2 and SO2) compared with conventional sputum aspiration, which highlights the importance of flexible bronchoscopes in airway management for SP patients undergoing IMV in the ICU. We also observed that the mean optimal frequency of flexible bronchoscopes affected the prognosis of such critically ill patients. It is our recommendation that the mean frequency of flexible bronchoscopes be ≤once every 3 days in that it contributed to better survival outcomes than the mean frequency of >once every 3 days. In addition, we concluded that more frequent B-ACT may be more suitable for improving prognosis in these patients with cardiopulmonary diseases or positive cultures for bacteria. Hopefully, our study could provide a novel evidence-based individualized therapeutic reference intervention for SP patients receiving with IMV in the ICU”. (please see line 384 in the modified version).

Reviewer 4 Report

 The aim of this study was  to investigate the effectiveness of bronchoscopic airway 11 clearance therapy (B-ACT) on severe pneumonia patients with invasive mechanical ventilation.

The results on survival and improvements in clinical and lab parameters after B-ACT were especially interesting and important.

Overall, the study was well-conducted, the data analysis was appropiate, and the results were well-presented.

I have a few comments for some minor typographical errors and to improve readability. 

Line 24:  play misspelled

Line 54:   better: However, the effectiveness of B-ACT in SP patients during IMV has remained controversial over the years, knowing that with B-ACT recommendations in currently available guidelines mainly relying on studies enrolling SP patients without undergoing not receiving IMV

Line 74: capitalize We

Indicate the source of the artificial sputum.

Line 102:  Better: 100-250 mL normal saline

LINe 105:  I DID NOT UNDERSTAND THESE SENTENCES: Surgical records, medical records and instructions should be improved. The preoperative disease evaluation and postoperative close observation were also important to the patients.

LINE 118-119: PLEASE DEFINE WHAT THESE ARE: The laboratory results included routine blood parameters, blood gas analysis, inflammatory factors, blood biochemical parameters, and myocardium markers.

Figure 3.  Legend is too small

Line 297:  delete space, insert space: interleukin-8 (IL-8)

Line 280:  better: Our analysis showed that B-ACT could improves some important laboratory results parameters, including levels of inflammatory biomarkers (CRP and PCT), leukocytes (WBC# and N%), PO2,  and SO2.

Line 284: tumor necrosis factor-alpha (TNF-α),

Line 297:  delete space, insert space: interleukin-8 (IL-8)

Line 300:better:  rolled 72 SP patients including those receiving with IMV concluded…

Line 303:  better:  in to assess the impact of B-ACT on the different laboratory results  parameters in those patients after

Line 313-314:  better: Our study found that the mean bronchoscopy frequency of ≤once every 3 days improves survival to hospital discharge compared to >once every 3 days for SP patients during IMV.

Line 324: better: and found that early BAL under bronchoscopy could did not improve the survival rate but can reduced the duration of hospitalization and ICU time, …

Line 345:  Better: Therefore, more further researches are is needed to investigate the impacts of B-ACT on these critically-ill patients of different genders, ages, regions, or other clinical characteristics. suggesting great value in clinical application

Line 381: patients with receiving IMV in the future

Line 384: better: In this study, we demonstrated that B-ACT significantly prolonged the time of improved sur- vival to hospital discharge and improved many different laboratory parameters, which highlights the importance of flexible bronchoscopes in airway management for SP patients undergoing IMV. in the ICU

Line 393: better: novel evidence-based individualized therapeutic reference intervention for SP patients receiving with IMV.

no

Author Response

Point 1: Line 24:  play misspelled

Response 1: Thank you for your constructive suggestions. The advice you raised is very meaningful. We modified the content according to your requirement (please see line 24 in the modified version).

Point 2: Line 54:   better: However, the effectiveness of B-ACT in SP patients during IMV has remained controversial over the years, knowing that with B-ACT recommendations in currently available guidelines mainly relying on studies enrolling SP patients without undergoing not receiving IMV

Response 2: Thank you for your constructive suggestions. The advice you raised is very meaningful. We modified the content according to your requirement (please see line 54 in the modified version).

Point 3: Line 74: capitalize We

 Indicate the source of the artificial sputum.

 Response 3: Thank you for your constructive suggestions. The advice you raised is very meaningful. We modified the content according to your requirement (please see line 74 in the modified version). We added the different criteria for B-ACT and artificial sputum (please see line 77 in the modified version).

 Point 4: Line 102:  Better: 100-250 mL normal saline

 Response 4: Thank you for your constructive suggestions. The advice you raised is very meaningful. We modified the content according to your requirement (please see line 102 in the modified version).

Point 5: LINe 105:  I DID NOT UNDERSTAND THESE SENTENCES: Surgical records, medical records and instructions should be improved. The preoperative disease evaluation and postoperative close observation were also important to the patients.

Response 5: Thank you for your constructive suggestions. The advice you raised is very meaningful. Before B-ACT, doctors and nurses should complete the preoperative disease evaluation and give medical orders or instructions to patients. After B-ACT, doctors and nurses should complete postoperative close observation and write related bronchoscopic records.

Point 6: LINE 118-119: PLEASE DEFINE WHAT THESE ARE: The laboratory results included routine blood parameters, blood gas analysis, inflammatory factors, blood biochemical parameters, and myocardium markers.

Response 6: Thank you for your constructive suggestions. The advice you raised is very meaningful. We added the corresponding definition and modified it as “The laboratory results included routine blood parameters (PLT, WBC, N%), blood gas analysis (PH, PCO2, PO2, SO2), inflammatory factors (PCT, CRP), blood biochemical parameters (Glu, ALT, AST, Urea, Cr, Alb), and myocardium markers (TnI, BNP, D-dimer) (please see line 118-119 in the modified version).

Point 7: Figure 3.  Legend is too small

Response 7: Thank you for your constructive suggestions. The advice you raised is very meaningful. We modified the figure legend (please see modified figure 3 in the modified version).

Point 8: Line 297:  delete space, insert space: interleukin-8 (IL-8)

Responses 8: Thank you for your constructive suggestions. The advice you raised is very meaningful. We modified the content according to your requirement (please see line 297 in the modified version).

Point 9: Line 280:  better: Our analysis showed that B-ACT could improves some important laboratory results parameters, including levels of inflammatory biomarkers (CRP and PCT), leukocytes (WBC# and N%), PO2, and SO2.

Response 9: Thank you for your constructive suggestions. The advice you raised is very meaningful. We modified the content according to your requirement (please see line 280 in the modified version).

Point 10: Line 284: tumor necrosis factor-alpha (TNF-α),

Response 10: Thank you for your constructive suggestions. The advice you raised is very meaningful. We modified the content according to your requirement (please see line 284-285 in the modified version).

Point 11: Line 297:  delete space, insert space: interleukin-8 (IL-8)

Response 11: Thank you for your constructive suggestions. The advice you raised is very meaningful. We modified the content according to your requirement (please see line 297 in the modified version).

Point 12: Line 300: better:  rolled 72 SP patients including those receiving with IMV concluded…

Response 12: Thank you for your constructive suggestions. The advice you raised is very meaningful. We modified the content according to your requirement (please see line 300 in the modified version).

Point 13: Line 303:  better:  in to assess the impact of B-ACT on the different laboratory results parameters in those patients after

Response 12: Thank you for your constructive suggestions. The advice you raised is very meaningful. We modified the content according to your requirement (please see line 303 in the modified version).

Point 14: Line 313-314:  better: Our study found that the mean bronchoscopy frequency of ≤once every 3 days improves survival to hospital discharge compared to >once every 3 days for SP patients during IMV.

Response 14: Thank you for your constructive suggestions. The advice you raised is very meaningful. We modified the content according to your requirement (please see line 313-314 in the modified version).

Point 15: Line 324: better: and found that early BAL under bronchoscopy could did not improve the survival rate but can reduced the duration of hospitalization and ICU time, …

Response 15: Thank you for your constructive suggestions. The advice you raised is very meaningful. We modified the content according to your requirement (please see line 324 in the modified version).

Response 16: Line 345:  Better: Therefore, more further researches are is needed to investigate the impacts of B-ACT on these critically-ill patients of different genders, ages, regions, or other clinical characteristics. suggesting great value in clinical application

Response 16: Thank you for your constructive suggestions. The advice you raised is very meaningful. We modified the content according to your requirement (please see line 345 in the modified version).

Point 17: Line 381: patients with receiving IMV in the future

Response 17: Thank you for your constructive suggestions. The advice you raised is very meaningful. We modified the content according to your requirement (please see line 381 in the modified version).

Point 18: Line 384: better: In this study, we demonstrated that B-ACT significantly prolonged the time of improved survival to hospital discharge and improved many different laboratory parameters, which highlights the importance of flexible bronchoscopes in airway management for SP patients undergoing IMV. in the ICU

Response 18: Thank you for your constructive suggestions. The advice you raised is very meaningful. We modified the content according to requirements of you and reviewer 3 (please see line 384 in the modified version). Reviewer 3 suggested that the conclusion is too general.

Point 19: Line 393: better: novel evidence-based individualized therapeutic reference intervention for SP patients receiving with IMV.

Response 19: Thank you for your constructive suggestions. The advice you raised is very meaningful. We modified the content according to requirements of you and reviewer 3 (please see line 393 in the modified version). Reviewer 3 suggested that the conclusion is too general.

Reviewer 5 Report

Dear Authors,

I read your manuscript with interest.

The management of the intubated patient with infectious airway disease remains a high-stakes problem. the diatribe between those who consider continuous aspiration of secretions detrimental and those who propose seriate toilets is still alive.

Although affected by numerous limitations, which you have skillfully pointed out in the text, I believe that the results and case history of your study are important in defining further scientific guidance on the topic.

I only ask you to specify whether the disposable bronchoscopes used were changed each time the procedure was performed (even for the same patient). And whether in cases where multi-use bronchoscopes were utlized, a proper sterilization process was performed.

Congratulations.

Minor spelling, comas, and prepostions used.

Author Response

Response: Thank you very much for your affirmation. In this study, we confirmed that the single-use flexible bronchoscopes (SUFB) used were changed each time when the procedure was performed even for the same patient. And all the reusable flexible bronchoscopes (RFB) were utilized by a proper sterilization process after each use. We have reviewed all the requirements and confirmed that our revisions meet the requirements of the magazine. Thanks again for your efforts on our work.

Round 2

Reviewer 3 Report

Congratulations for your work!